# A Nationwide Population-Based Study on the Association between Land Transport Accident and Peripheral Vestibular Disorders

**DOI:** 10.3390/ijerph18126570

**Published:** 2021-06-18

**Authors:** Herng-Ching Lin, Sudha Xirasagar, Chia-Hui Wang, Yen-Fu Cheng, Tsai-Ching Liu, Tzong-Hann Yang

**Affiliations:** 1School of Health Care Administration, College of Management, Taipei Medical University, Taipei 110, Taiwan; henry11111@tmu.edu.tw; 2Sleep Research Center, Taipei Medical University Hospital, Taipei 110, Taiwan; 3Department of Health Services Policy and Management, Arnold School of Public Health, University of South Carolina, Columbia, SC 29210, USA; sxirasagar@sc.edu; 4Department of Urban Development, University of Taipei, Taipei 110, Taiwan; chwang@utaipei.edu.tw; 5Research Center of Sleep Medicine, College of Medicine, Taipei Medical University, Taipei 110, Taiwan; entist@gmail.com; 6Department of Otolaryngology-Head and Neck Surgery, Taipei Veterans General Hospital, Taipei 112, Taiwan; 7Department of Speech, Language and Audiology, National Taipei University of Nursing and Health, Taipei 112, Taiwan; 8Department of Public Finance, Public Finance and Finance Research Center, National Taipei University, New Taipei City 237, Taiwan; tching@mail.ntpu.edu.tw; 9Department of Otorhinolaryngology, Taipei City Hospital, Taipei 110, Taiwan

**Keywords:** peripheral vestibular disorders, land transport accidents, epidemiology

## Abstract

This case–control study aimed to investigate the association of peripheral vestibular disorders (PVD) with subsequent land transport accidents. Data for this study were obtained from Taiwan’s National Health Insurance (NHI) dataset. We retrieved 8704 subjects who were newly found to have land transport accidents as cases. Their diagnosis date was used as their index date. Controls were identified by propensity score matching (one per case, *n* = 8704 controls) from the NHI dataset with their index date being the date of their first health service claim in 2017. Multiple logistic regressions were performed to calculate the prior PVD odds ratio of cases vs. controls. We found that 2.36% of the sampled patients had been diagnosed with PVD before the index date, 3.37% among cases and 1.36% among controls. Chi-square test revealed that there was a significant association between land transport accident and PVD (*p* < 0.001). Furthermore, multiple logistic regression analysis suggested that cases were more likely to have had a prior PVD diagnosis when compared to controls (OR = 2.533; 95% CI = 2.041–3.143; *p* < 0.001). After adjusting for age, gender, hypertension, diabetes, coronary heart disease, and hyperlipidemia, cases had a greater tendency to have a prior diagnosis of PVD than controls (OR = 3.001, 95% CI = 2.410–3.741, *p* < 0.001). We conclude that patients with PVD are at twofold higher odds for land transport accidents.

## 1. Introduction

Peripheral vestibular disorders (PVD) are very common in clinic settings [1] with an estimated one-year prevalence of 1.2–6.5% in the adult population [2]. The disease entity comes from mismatched sensations of vestibular afferents, the visual system, and somatosensory and proprioceptive receptors with a plethora of pathogenesis, including Meniere’s disease (MD), benign paroxysmal positional vertigo (BPPV), vestibular neuritis (VN), and other or unspecified peripheral vestibular dizziness. Vestibular function is also involved in spatial orientation and navigation [3,4]. Patients with PVD often experience vertigo as a hallucination of motion; some describe this as self-motion and others as motion of the environment. Common symptoms include spinning sensation, postural and gait disturbance, swaying or tilting, sudden fall, nausea and vomiting, oscillopsia, drop attacks, and spatial disorientation, particularly when turning and driving on open, featureless roads. This could possibly limit the ability to drive [5,6] and might therefore increase the risk of land transport accidents.

Land transport accidents indicate injuries related to the flow of vehicles and people on public roads, delineated by the classification from the International Classification of Diseases, 10th Revision (ICD-10). Land transport accidents, also called traffic accidents, motor vehicle accidents, automobile accidents, or car accidents, occur when a vehicle collides with another vehicle, pedestrian, animal, road debris, or other stationary objects, such as a tree, pole, or building. Land transport accidents often cause injury, disability, death, and property damage as well as financial costs to individuals, their families, and to the nation as a whole, up to three percent of gross domestic product. Land transport accidents are an important and prevailing public health issue in Taiwan, having caused 17,915 deaths in Taiwan between 2011 and 2020 [7]. Furthermore, the injury rate of land transport accidents has risen rapidly in Taiwan [7]. The World Health Organization (WHO) also claimed a rise in road traffic deaths of more than 1.35 million worldwide annually, and the 2030 Agenda for Sustainable Development has set an ambitious target of halving the global number of deaths and injuries from road traffic crashes by 2020 [8].

Patients with PVD have reported driving difficulty in several studies. For example, one hospital-based retrospective study among Omani drivers found that approximately 25% of victims had prior symptoms of dizziness. In addition, vertigo was felt by some drivers prior to the accident [9]. Another annual nationwide survey of a noninstitutionalized civilian population showed about 44% of patients with bilateral vestibulopathy either stopped driving or changed driving habits as a result of their dizziness. They also reported that the majority of respondents with bilateral vestibulopathy had motion discomfort, particularly when riding as a passenger in a car, bus, train, or plane [10]. Furthermore, one study using data from the 2016 National Health Interview Survey of U.S. adults found that patients with vestibular vertigo were reported to have over threefold higher odds (odds ratio, 3.5; 95% confidence interval, 1.7–7.3) of automobile accidents [11]. However, although the above studies reported an association between vestibular dysfunction and increased risk of motor vehicle accidents, there is limited population-based epidemiological evidence of the risk of land transport accidents following a PVD diagnosis. Additionally, the American Medical Association (AMA) calls for physicians to identify patient’s driving impairments, encourages safe driving, and advises doctors to require patient driving evaluation by state licensing agencies when necessary [12]. However, driving restrictions for patients with vertigo and dizziness differ greatly from country to country [13]. Taiwan, for example, makes no specific binding requirements, whereas Germany has enforced very strict regulations since 2014 [13]. These differences reflect the lack of population-based studies on the risk of land transport accidents in patients with a vestibular disease in each country, especially Taiwan.

Therefore, to develop strategies to solve injuries from road traffic accidents in Taiwan, this study aimed to determine the risk of land transport accident occurrence among a population-wide cohort of patients diagnosed with PVD using a nationwide population-based retrospective case–control study.

## 2. Methods

### 2.1. Database

The data for this case–control study were retrieved from Taiwan’s National Health Insurance dataset (NHIRD). The NHIRD, which is administrative data from the Taiwan National Health Insurance (NHI) program, includes claims data and registration files of over 99% of all Taiwanese residents (*n* = 23.72 million). Claims submitted by healthcare providers are accompanied by data, including diagnostic codes, procedure codes, prescription drugs, and direct medical costs of both inpatient and ambulatory care. The Bureau of the National Health Insurance collects data from the NHI program annually and sorts it into registration files and original claim data for reimbursement. The Bureau of the National Health Insurance deidentifies these data files by scrambling the identification codes of both patients and medical facilities. Thereafter, these data are sent to the National Health Research Institutes to form NHIRD files. In addition, the NHIRD database includes a registry of contracted medical facilities, a registry of board-certified surgeons, a monthly claims summary for in-patient claims, and details of in-patient orders and expenditure on prescriptions dispensed at contracted pharmacies. It also provides principal operational procedures along with one principal diagnosis code and up to four secondary diagnosis codes for each patient from the International Classification of Disease, Ninth Revision, Clinical Modification (ICD-9-CM).

The NHIRD has been used by many researchers in Taiwan to conduct clinical–epidemiological studies that have been published in internationally peer-reviewed journals. Therefore, the NHIRD provides an excellent opportunity to investigate the association of land transport accident with prior vertigo using nationwide population-based data.

The Institutional Review Board of Taipei Medical University approved this study (TMU-JIRB N202102033). This study adheres to the STROBE guidelines for research reporting standards and is compliant with the Declaration of Helsinki.

### 2.2. Selection of Cases and Controls

For case selection, we identified 8704 patients aged between 20–79 years who had received a diagnosis of land transport accident (ICD-10-CM code V01-V89) in ambulatory care or emergency departments between January and December 2017. We further defined the diagnosis date of the first claim for transport accident as the index date for cases.

We likewise retrieved controls from the NHIRD 2017 registry of beneficiary. First, we included only those enrollees aged between 20 and 79 years who did not have a history of land transport accident. Thereafter, propensity score matching was used to select 8704 controls by matching cases in terms of demographic variables (age, sex, monthly income (TWD 0–15,840, TWD 15,841–25,000, and ≥TWD 25,001; average exchange rate in 2017: USD 1.00 = TWD 30), geographic location (northern, central, southern, and eastern) and urbanization level of the patient’s residence (levels 1–5, with 1 indicating the most urbanized and 5 indicating the least urbanized), and medical comorbidities including hyperlipidemia (ICD-9-CM code 272.4 or ICD-10-CM codes E78.5), diabetes (ICD-9-CM code 250 or ICD-10-CM codes E08–E13), coronary heart disease (ICD-9-CM codes 411–414 or ICD-10-CM codes I20–I25), and hypertension (ICD-9-CM codes 401–405 or ICD-10-CM code I10). For the classification of urbanization level, all 359 cities/towns in Taiwan were stratified into five levels based on a composite score obtained by calculating population density (people/km^2^), population ratio of people with an educational level of college or above (%), ratio of people over 65 years (%), ratio of agriculture workers (%), and the number of physicians per 100,000 people.

We calculated a propensity score for each patient. The propensity score was initially used to balance demographic and medical comorbidities, which were distributed unequally between enrollees who had received a diagnosis of land transport accident and those who had not. Because the probability of receiving a diagnosis of land transport accident may depend on the enrollee’s demographic and medical comorbidities, the variables of age, sex, monthly income, geographic location and urbanization level of the patient’s residence, hyperlipidemia, diabetes, coronary heart disease, and hypertension were entered into a multivariable logistic regression model as predictors to calculate the expected probability of receiving a diagnosis of land transport accident for each enrollee. Furthermore, all enrollees were grouped into deciles based on their propensity score. The model was then stratified by propensity score in deciles to ascertain that, within each stratum, controls were made for cases with a similar expected probability of receiving a diagnosis of land transport accident and, to a large extent, a similar distribution of confounders.

We further assigned the date of the control’s first utilization of ambulatory care in 2017 as their index date.

### 2.3. Exposure Assessment

We identified PVD cases based on ICD-9-CM codes 386.0, 386.1, 386.10, and 386.19 or ICD-10-CM codes H81.0, H81.1, H81.2, H81.3, H81.8, H81.9, H82, or H83 present in at least one claim prior to the index date. In addition, we only included PVD cases who had received at least one of the vertigo diagnoses made by an otorhinolaryngologist.

### 2.4. Statistical Analysis

The statistical analyses were performed using the SAS system (SAS System for Windows, version 8.2, SAS Institute Inc., Cary, NC, USA). Descriptive analyses, including frequency, percentage, mean, and standard deviation, were performed on all of the identified variables. First, we used chi-square tests to examine differences in the distribution of patients who had received a diagnosis of land transport accident and controls in terms of demographic variables and medical comorbidities. We further used logistic regression analysis to examine the association of land transport accident with prior PVD after adjusting for age, gender, monthly income, geographic location, urbanization level, hypertension, diabetes, coronary heart disease, and hyperlipidemia. We used a conventional two-tailed value of *p* < 0.05 to assess statistical significance.

## 3. Results

Table 1 presents the data on demographic variables and medical comorbidities for 8704 patients who had received a diagnosis of land transport accident and 8704 controls. We found that after using propensity score matching, there were significant differences between patients who had received a diagnosis of land transport accident and controls regarding age (*p* < 0.001), gender (*p =* 0.025), monthly income (*p* < 0.001), geographic location (*p* < 0.001), and residential urbanization level (*p* < 0.001). Because we used propensity score matched controls, it was expected that we would find small but statistically significant differences in the variables used to match controls with patients who had received a diagnosis of land transport accident. The mean age was 43.21 and 45.68 years for patients who had received a diagnosis of land transport accident and controls, respectively. The majority of the sampled patients resided in northern Taiwan, and only about 16.4% resided in southern Taiwan. The distribution of the level of urbanization indicated that the majority of the study sample was found within communities situated in urbanization level 2. In addition, we found that controls had significantly higher prevalence of comorbidities than patients who had received a diagnosis of land transport accident in terms of hyperlipidemia (26.30% vs. 19.84%, *p* < 0.001), diabetes (17.98% vs. 13.67%, *p* < 0.001), hypertension (27.77% vs. 22.07%, *p* < 0.001), and coronary heart disease (11.09% vs. 7.16%, *p* < 0.001).

The prevalence of prior PVD in the study sample is presented in Table 2. We found that 2.36% of the sampled patients had been diagnosed with PVD before the index date, 3.37% among patients who had received a diagnosis of land transport accident and 1.36% among controls. Chi-square test revealed that there was a significant association between land transport accident and PVD (*p* < 0.001). Furthermore, multiple logistic regression analysis suggested that patients who had received a diagnosis of land transport accident were more likely to have had a prior PVD diagnosis when compared to controls (OR = 2.533; 95% CI = 2.041–3.143; *p* < 0.001).

Table 3 shows the covariate-adjusted OR for land transport accident among the sampled patients. It can be seen that after adjusting for age, gender, hypertension, diabetes, coronary heart disease, and hyperlipidemia, patients who had received a diagnosis of land transport accident had a greater tendency to have a prior diagnosis of PVD than controls (OR = 3.001, 95% CI = 2.410–3.741, *p* < 0.001). In addition, we found that age was negatively associated with land transport accident (adjusted OR = 0.996, 95% CI = 0.994–0.998, *p* < 0.001) after adjusting for gender, hypertension, diabetes, coronary heart disease, and hyperlipidemia. Furthermore, there was no significant association between gender and land transport accident (adjusted OR = 1.044, 95% CI = 0.982–1.109, *p* = 0.169) after adjusting for age, hypertension, diabetes, coronary heart disease, and hyperlipidemia.

## 4. Discussion

In this first population-based study of Taiwan’s adult population from NHIRD, subjects with prior PVD were found to have over twofold higher odds of land transport accidents than individuals matched to them on sociodemographic and medical comorbidity status. The strong association of prior PVD with land transport accident reveals the necessity for physicians to counsel patients with vertigo and draft extensive evaluations and advice on driving fitness.

As shown in several studies that parallel our research, vertigo/dizziness is one of the causes of road traffic accidents. Three potential explanations might account for impaired driving fitness after a diagnosis of vestibular vertigo. The first possible reason for the increase in hazards may be spatial disorientation, especially at higher speeds and on curved roads [5] and attacks of episodic vertigo without prodromal symptoms [13]. Because driving a motor vehicle requires visuospatial ability and vestibular function [14,15], patients with vestibular vertigo are likely to face challenges during situations such as driving in the rain, driving alone, making left turns across traffic, freeway driving, driving in a local road with high traffic, rush hour driving, night driving, in parking spaces, changing lanes, staying in lane, traffic checks, and driving in ramped garages, which are all conditions with limited visual input [6]. Consequently, they may pull off the road [15]. In another hospital-based survey, approximately 25% of victims had prior symptoms of dizziness [9]. Another study using data from the 2008 United States National Health Interview Survey showed about 44% of patients with bilateral vestibulopathy either stopped driving or changed their driving habits as a result of their dizziness [10]. Similar to our study, in a recent population-based cross-sectional study using data from the 2016 United States National Health Interview Survey, patients with vestibular vertigo were reported to have over threefold higher odds of automobile accidents [11]. The second possible explanation for higher risk of land transport accidents may be acute rotatory vertigo attack during the drive. Drop attack, a sudden loss of tone mediated by vestibulospinal reflexes, may occur in patients with MD [16], superior canal dehiscence [17], and aminoglycoside toxicity [18]. Motor vehicle accidents may be a consequence of an acute episode of vertigo evoked by head movements in drivers with BPPV [13]. Moreover, all vertigos are made worse by movement of the head, which is essential in driving. The third possible cause for increased risk of land transport accidents may be the use of vestibular suppressant medications, such as antiemetics, antihistamines, and benzodiazepines [19], in drivers with vestibular vertigo. These medicines are considered driving-impairing medicines [20,21] because of their adverse effects, including dizziness, drowsiness, confusion, fatigue, blurred vision, agitation, akathisia, tardive dyskinesia, dystonia, amnesia, and even vertigo. Driving a motor vehicle is a multifaceted task and demands proper cognitive and psychomotor skills, such as concentration, alertness, response time, visual acuity, dexterity, and coordination. [22] Medications can adversely influence these driving-related skills and consequently post a threat to traffic safety [23]. Therefore, in settings with alternative transportation available, physicians may suggest vertigo patients to avoid driving. Legislation to prohibit driving by certain vestibular vertigo patients who are not fit to drive could be set up, although it varies widely across countries [13,24].

However, inconsistent findings have also been observed in the literature. An example is how certain vestibular diseases (MD and VN) might have led to whiplash injuries (a surrogate marker of these conditions in traffic accidents). Although the accident rates increased, they were not different before and after the diagnosis of vestibular diseases [25]. In another study, the self-reported crash rate and rate of citations for moving violations did not differ between normal subjects and vertigo patients, including those with BPPV, MD, chronic vestibulopathy, postoperative acoustic neuroma resections, and postoperative vestibular nerve section [6]. Recently, a questionnaire-based study in Finland concluded that patients with MD had fewer episodes of traffic accidents when compared to the general population. This correlation may be traced to the fact that patients with MD practice more caution in making decisions regarding when and when not to drive [26]. Moreover, one study reported that there was no significant difference in the risk of injuries in the next year between patients with vertigo who visited the emergency department and patients with renal colic [27]. Having alternative methods of transportation other than personal driving may explain why differences exist in findings on subsequent injury risk in different settings.

The present study has numerous strengths. Since the initiation of Taiwan’s NHI in 1995, the healthcare system in Taiwan has become easily accessible and financially affordable with extremely low copayments for each resident. This has reduced the possibility of selection bias that may result from discrepancies in socioeconomic status or residential location. In addition, we were able to gather data on diagnoses of prior PVD and land transport accidents from all sources as the NHIRD covers every utilization of medical care by all Taiwanese residents (about 23 million), with records comprising ambulatory care visits, emergency department visits, inpatient admissions, etc. Having comprehensive, affordable, and accessible healthcare provided by the NHI program has enabled patients with alarming medical conditions, such as vertigo and transport accidents, to seek immediate medical visit. Consequently, socioeconomic status would not be a factor that differentiates the identification of vertigo and transport accidents. The employment of NHI using claims data further avoids possible recall bias that are typically related to self-reported survey data. The present study was designed as a case–control study that selected controls by propensity score matching, which ultimately strengthens the validity of the findings, minimizes selection bias and misclassification bias, and supports causal inferences between vestibular vertigo and land transport accidents.

Despite the abovementioned strengths, there are some study limitations. NHIRD claims lack certain critical items of data, such as visual acuity and visual field, cause of the land transport accidents, and victim categories (driver, passenger, or pedestrian). Another limitation is the lack of data on factors relating to the environment, living habits, risk-taking behavior, and genetic parameters. Furthermore, given that this study may lack the inclusion of some land transport accidents that were noninjurious and thus unreported, the results from this study should be analyzed with caution. Despite being a population-based study, our findings may not be generalized to other regions or countries due to the different factors of ethnicity and living environment. Finally, the model utilized in this study did not take coprescribed medications into consideration. Several medications, such as antidepressants and other psychoactive medications, may increase the risk of land transport accidents.

## 5. Conclusions

In conclusion, our analysis quantifies that subjects with prior diagnosis of peripheral vestibular disorders have over twofold higher odds of land transport accidents than controls in Taiwan. It may guide physicians as they counsel patients with vestibular impairment on possible safety threats of driving a motor vehicle. Careful assessment and precaution are suggested to prevent land transport accidents in this population. Further, the findings may guide future settings and enforcement laws relating to vestibular vertigo and raise public awareness about this issue.

## Figures and Tables

**Table 1 ijerph-18-06570-t001:** Demographic characteristics of patients diagnosed with transport accident during 2017 and control patients in Taiwan (*n* = 17,408).

Variable	Patients with Transport Accident (*n* = 8704)		Controls (*n* = 8704)		*p* Value
	Total No.	%	Total No.	%	
Age, mean (SD)	43.21	(17.44)	45.68	(15.95)	<0.001
Gender					
Male	4159	47.78%	4307	49.48%	0.025
Female	4545	52.22%	4397	50.52%	
Hyperlipidemia	1727	19.84%	2289	26.30%	<0.001
Diabetes	1190	13.67%	1565	17.98%	<0.001
Hypertension	1921	22.07%	2417	27.77%	<0.001
Coronary heart disease	623	7.16%	965	11.09%	<0.001

**Table 2 ijerph-18-06570-t002:** Prevalence of vertigo and crude odds ratio of prior peripheral vestibular disorders among cases vs. controls.

Presence of Prior Peripheral Vestibular Disorders	Total (*n* = 17,408)		Patients with Transport Accident (*n* = 8704)		Controls (*n* = 8704)	
	*n*, %		*n*, %		*n*, %	
Yes	411	2.36%	293	3.37%	118	1.36%
No	16997	97.64%	8411	96.63%	8586	98.64%
OR (95% CI)	--		2.533 (2.041–3.143)		1.00	

Notes: OR = odds ratio

**Table 3 ijerph-18-06570-t003:** Covariate-adjusted odds of prior peripheral vestibular disorders among the sampled patients (*n* = 17,408).

Variables	Presence of Transport Accident		
	Odds Ratio	95% CI	*p* Value
Prior peripheral vestibular disorders	3.001	(2.410–3.741)	<0.001
Age	0.996	(0.994–0.998)	<0.001
Female	1.044	(0.982–1.109)	0.169
Hyperlipidemia	0.798	(0.732–0.87)	<0.001
Diabetes	0.931	(0.845–1.024)	0.140
Hypertension	0.938	(0.862–1.021)	0.140
Coronary heart disease	0.724	(0.646–0.811)	<0.001

Notes: CI = confidence interval.

## Data Availability

The National Health Insurance Research Database, which has been transferred to the Health and Welfare Data Science Center (HWDC). Interested researchers can obtain the data through formal application to the HWDC, Department of Statistics, Ministry of Health and Welfare, Taiwan (http://dep.mohw.gov.tw/DOS/np-2497-113.html, accessed on 2 June 2021).

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
