# Peer review of "A Nationwide Population-Based Study on the Association between Land Transport Accident and Peripheral Vestibular Disorders"

_ijerph, 2021, doi:10.3390/ijerph18126570_

Round 1

Reviewer 1 Report

A nationwide population-based study on the association between land transport accident and peripheral vestibular disorders

I thank you for the opportunity to comment this study. Topic is important but there are relatively many studies already now about this research subject.

Main result

After adjusting for age, gender, monthly income, geographic location, urbanization level, hypertension, diabetes, coronary heart disease and hyperlipidemia, cases had a greater tendency to have a prior diagnosis of PVD than controls (OR 2.993, 95% 2.402-3.731).

Comments

  1. In general, I found writing style unfinished. Quite rarely you see manuscripts that have not followed in the text the journal’s citation guidelines. Reading this manuscript was substantially hampered by this fact. This comment applies partly also how the references were formatted.
  2. Please, see the reference below which defines which diagnosis belong to peripheral vestibular diseases. I don’t understand why f.ex. Meniere’s disease was not included this study. Authors need to add also patients having diagnosis H81.0 to the study population.

Strupp M, Brandt T. Peripheral vestibular disorders. Curr Opin Neurol. 2013 Feb;26(1):81-9. doi: 10.1097/WCO.0b013e32835c5fd4. PMID: 23254559.

  1. Dizziness is not a persistent symptom and some of the diseases diagnosed hardly cause any more symptoms in patients, it is necessary to analyze a group of patients with a recent diagnosis related to dizziness. I would suggest a pool of patients with dizziness within a year.
  2. Additionally, analysis separately patients who have diagnosis H81.0 (menires’s disease) or vestibular neuritis (H81.2) using current analytics and also one year time point.
  3. When authors carry out adjustments, they must justify why these adjustments are carried out. F. ex. if authors believe that income or cholesterol level is important to adjust, they must justify this. This applies all adjustments.
  4. Table 1 & 3 contain much unnecessary data. Please, simplify. Clarify table 2.
  5. There is no data available of accident types. This needs to be presented. Are ceratin types of accidents potentially related to dizziness?
  6. Clarify and sharpen bot the introduction.
  7. Authors assume on discussion that dizziness is related to these accidents because of previous diagnosis of peripheral vestibular disorders. Please, provide data regarding different types of vestibular disorders how often there are symptoms and whether there are, for example, pre-symptoms (Meniere's disease), in which case the driver may avoid driving.
  8. Are the authors aware of any recommendations how a doctor can assess the symptom of dizziness at a driving license check.

Author Response

Reviewer 1

A nationwide population-based study on the association between land transport accident and peripheral vestibular disorders

 I thank you for the opportunity to comment this study. Topic is important but there are relatively many studies already now about this research subject.

 Main result

After adjusting for age, gender, monthly income, geographic location, urbanization level, hypertension, diabetes, coronary heart disease and hyperlipidemia, cases had a greater tendency to have a prior diagnosis of PVD than controls (OR 2.993, 95% 2.402-3.731).

 Comments

  1. In general, I found writing style unfinished. Quite rarely you see manuscripts that have not followed in the text the journal’s citation guidelines. Reading this manuscript was substantially hampered by this fact. This comment applies partly also how the references were formatted.

Response: Indeed, the format for this manuscript was prepared by the Journal. The Journal revises the manuscript format including references before sending out for review.   

  1. Please, see the reference below which defines which diagnosis belong to peripheral vestibular diseases. I don’t understand why f.ex. Meniere’s disease was not included this study. Authors need to add also patients having diagnosis 0 to the study population.

Strupp M, Brandt T. Peripheral vestibular disorders. Curr Opin Neurol. 2013 0.dizziness within a year.

Response: We have re-analyzed our data and found that about 91% of PVD cases had ever received a PVD diagnosis within one year prior to the index date. The association between land transport accident and prior PVD still exists even only including those PVD cases who had ever received a PVD diagnosis within one year prior to the index date.    

  1. Additionally, analysis separately patients who have diagnosis H81.0 (menires’s disease) or vestibular neuritis (H81.2) using current analytics and also one year time point.

Response: We found that only 8 and 7 MD cases in cases and controls, respectively,  The case number for MD was not large enough to perform statistical analysis. We are sorry for this!

  1. When authors carry out adjustments, they must justify why these adjustments are carried out. F. ex. if authors believe that income or cholesterol level is important to adjust, they must justify this. This applies all adjustments.

Response: We adjusted for socio-demographic characteristics and common medical comorbidities. The reason for the adjustment of these common medical comorbidities is that the medications for the treatment of these medical comorbidities may increase the risk of traffic accidents. In addition, age and sex may increase the risk of traffic accidents. We have removed income, urbanization level, and geographic region from the text.   

  1. Table 1 & 3 contain much unnecessary data. Please, simplify. Clarify table 2.

Response: We have removed the variables of income, urbanization level, and geographic region from the Tables 1 & 3. The following statements in Results have clearly explained Table 2 (The prevalence of prior PVD on the study sample was presented in Table 2. We found that 2.36% of the sampled patients had been diagnosed with PVD before the index date, 3.37% among patients who had received a diagnosis of land transport accident and 1.36% among controls. Chi-square test reveals that there was a significant association between land transport accident and PVD (p<0.001). Furthermore, multiple logistic regression analysis suggests that patients who had received a diagnosis of land transport accident were more likely to have had a prior PVD diagnosis when compared to controls (OR=2.533; 95% CI=2.041-3.143; p<0.001).)

  1. There is no data available of accident types. This needs to be presented. Are ceratin types of accidents potentially related to dizziness?

Response: According to the ICD-10-CM classifications, the accident types of “Land transport accident” includes the following 9 categories: Pedestrian injured in transport accident, Pedal cycle rider injured in transport accident, Motorcycle rider injured in transport accident, Occupant of three-wheeled motor vehicle injured in transport accident, Car occupant injured in transport accident, Occupant of pick-up truck or van injured in transport accident, Occupant of heavy transport vehicle injured in transport accident, Bus occupant injured in transport accident and Other land transport accidents. We did not have enough PVD case number to further analyze the association between PVD and land transport accident according to accident type. We are sorry for this!

  1. Clarify and sharpen bot the introduction.

Response: We do not understand your comment as it appears to be vague. We are sorry for this!

  1. Authors assume on discussion that dizziness is related to these accidents because of previous diagnosis of peripheral vestibular disorders. Please, provide data regarding different types of vestibular disorders how often there are symptoms and whether there are, for example, pre-symptoms (Meniere's disease), in which case the driver may avoid driving.

Response: As mentioned before, we did not have enough PVD case number to further analyze the association between PVD and land transport accident according to PVD type.

  1. Are the authors aware of any recommendations how a doctor can assess the symptom of dizziness at a driving license check.

Response: The physician in Taiwan would warn the patient not to drive during vertigo spell and/or on medication after the diagnosis of vertigo in the office visit. However, the physical check-up for vertigo prior issuing driver’s license is not include at the present time in Taiwan. The examinations in the check-up for driver’s license only include body height, weight, vision, color discrimination, hearing, limb mobility, visual field, and night vision.

Reviewer 2 Report

The manuscript aims to explore the association between traffic crashes and peripheral vestibular disorders (PDV) using data from Taiwan. The authors collected 8,704 PVD patients who involved in a traffic crash as the case group and the corresponding control group. From logistic regression modeling result, the patients with prior PVD are 2.993 times more likely to be involved in a crash after controlling other factors. The study’s topic is worthy of investigation and the manuscript is fairly written. However, the reviewer still has a few comments/suggestions on the manuscript.

The term, "land transport accidents" is unclear. The land transport systems include roads, urban metro transits (e.g., subway), intercity railways, etc. But from the manuscript, only traffic crash cases were used. The authors are strongly recommended to change the term.

The reviewer was confused because of the term “Victim (Line 224)”. The patients in the data are all victims? Anyone who is responsible for the crash occurrence has been deleted from the dataset? Please clarify.

Need to interpret the variables other than PVD (briefly). For example, why higher income people are more likely to be involved in a crash.

Please suggest specific policy implications based on the findings from the study.

There are many awkward English expressions in the manuscript.

Please consider applying a more advanced statistical approach such as random-parameter logit (or probit) modeling.

Author Response

Comments and Suggestions for Authors

The manuscript aims to explore the association between traffic crashes and peripheral vestibular disorders (PDV) using data from Taiwan. The authors collected 8,704 PVD patients who involved in a traffic crash as the case group and the corresponding control group. From logistic regression modeling result, the patients with prior PVD are 2.993 times more likely to be involved in a crash after controlling other factors. The study’s topic is worthy of investigation and the manuscript is fairly written. However, the reviewer still has a few comments/suggestions on the manuscript.

The term, "land transport accidents" is unclear. The land transport systems include roads, urban metro transits (e.g., subway), intercity railways, etc. But from the manuscript, only traffic crash cases were used. The authors are strongly recommended to change the term.

Response: Indeed, we included patients who had received a diagnosis of land transport accident (ICD-10-CM code V01-V89). According to the ICD-10-CM classifications, the cases with a diagnosis ICD-10-CM code V01-V89 were classified into the category of “land transport accidents”. The accident types of “Land transport accident” includes the following 9 categories: Pedestrian injured in transport accident, Pedal cycle rider injured in transport accident, Motorcycle rider injured in transport accident, Occupant of three-wheeled motor vehicle injured in transport accident, Car occupant injured in transport accident, Occupant of pick-up truck or van injured in transport accident, Occupant of heavy transport vehicle injured in transport accident, Bus occupant injured in transport accident and Other land transport accidents. We used the term “land transport accidents” in accordance with ICD-10-CM classifications.

The reviewer was confused because of the term “Victim (Line 224)”. The patients in the data are all victims? Anyone who is responsible for the crash occurrence has been deleted from the dataset? Please clarify.

Response: Thanks for your suggestion! We totally agree with your comments. We have changed “victim” to “subjects”.

Need to interpret the variables other than PVD (briefly). For example, why higher income people are more likely to be involved in a crash.

Response: As suggested by another reviewer, we have removed the variables of income, urbanization level, and geographic region from the text.

Please suggest specific policy implications based on the findings from the study.

Response: We have the following recommendations in the Conclusion: “In conclusion, our analysis quantifies that subjects with land transport accidents have over twofold increased odds of prior diagnosis of peripheral vestibular disorders than controls in Taiwan, and may guide physicians as they counsel patients with vestibular impairment on possible safety threat of driving a motor vehicle.”

There are many awkward English expressions in the manuscript.

Response: One of the authors is a native speaker. She has edited the whole manuscript. Thanks a lot!

Please consider applying a more advanced statistical approach such as random-parameter logit (or probit) modeling.

Response: Thanks for your suggestion. We think that logistic regression analysis can satisfactorily answer our question. We will consider random-parameter logit (or probit) modeling in the future studies.

Round 2

Reviewer 1 Report

I thank you for authors reply. However, I am confused that the authors do not want to carefully finalize the article.

This manuscript is a resubmission of an earlier submission. The following is a list of the peer review reports and author responses from that submission.